# Enhanced Photovoltaic Performance of Inverted Perovskite Solar Cells through Surface Modification of a NiO*_x_*-Based Hole-Transporting Layer with Quaternary Ammonium Halide–Containing Cellulose Derivatives

**DOI:** 10.3390/polym15020437

**Published:** 2023-01-13

**Authors:** I-Hsiu Ho, Yi-Jou Huang, Cheng-En Cai, Bo-Tau Liu, Tzong-Ming Wu, Rong-Ho Lee

**Affiliations:** 1Department of Chemical Engineering, National Chung Hsing University, Taichung 402, Taiwan; 2Department of Chemical and Materials Engineering, National Yunlin University of Science and Technology, Yunlin 64002, Taiwan; 3Department of Materials Science and Engineering, National Chung Hsing University, Taichung 402, Taiwan

**Keywords:** NiO*_x_*, quaternary ammonium halide, cellulose derivative, perovskite solar cells

## Abstract

In this study, we positioned three quaternary ammonium halide-containing cellulose derivatives (PQF, PQCl, PQBr) as interfacial modification layers between the nickel oxide (NiO*_x_*) and methylammonium lead iodide (MAPbI_3_) layers of inverted perovskite solar cells (PVSCs). Inserting PQCl between the NiO*_x_* and MAPbI_3_ layers improved the interfacial contact, promoted the crystal growth, and passivated the interface and crystal defects, thereby resulting in MAPbI_3_ layers having larger crystal grains, better crystal quality, and lower surface roughness. Accordingly, the photovoltaic (PV) properties of PVSCs fabricated with PQCl-modified NiO*_x_* layers were improved when compared with those of the pristine sample. Furthermore, the PV properties of the PQCl-based PVSCs were much better than those of their PQF- and PQBr-based counterparts. A PVSC fabricated with PQCl-modified NiO*_x_* (fluorine-doped tin oxide/NiO*_x_*/PQCl-0.05/MAPbI_3_/PC_61_BM/bathocuproine/Ag) exhibited the best PV performance, with a photoconversion efficiency (PCE) of 14.40%, an open-circuit voltage of 1.06 V, a short-circuit current density of 18.35 mA/cm^3^, and a fill factor of 74.0%. Moreover, the PV parameters of the PVSC incorporating the PQCl-modified NiO*_x_* were further enhanced when blending MAPbI_3_ with PQCl. We obtained a PCE of 16.53% for this MAPbI_3_:PQCl-based PVSC. This PQCl-based PVSC retained 80% of its initial PCE after 900 h of storage under ambient conditions (30 °C; 60% relative humidity).

## 1. Introduction

Organic–inorganic hybrid perovskite solar cells (PVSCs) have been attracting a tremendous amount of attention because of their outstanding photoconversion efficiencies (PCEs) and low production costs [1,2]. The PCEs and operational stabilities of PVSCs have improved dramatically within a very short period [3,4]. PVSCs fabricated from methylammonium lead halide perovskites (MAPbX_3_, where MA is a methylammonium (CH_3_NH_3_^+^) cation and X is a halide anion) are particularly interesting owing to their good optical absorption properties, high ambipolar charge transporting abilities, weakly bonded excitons that readily dissociate into free charges, and long electron–hole diffusion lengths [5,6,7,8]. In general, PVSCs can be divided into two classes depending on whether they have mesoporous or planar structures. Regular PVSCs with mesoporous structures possess a mesoporous metal oxide (TiO_2_) layer as the electron transporting layer (ETL), with the perovskite layer coated on top of a TiO_2_ layer presenting one of many tested hole transporting layers (HTLs) [9,10,11,12,13]. Although the highest PCEs have typically been produced from PVSCs having mesoscopic structures, the use of TiO_2_ is considered to be a disadvantage because it complicates the device preparation and weakens the stability of the cell under UV illumination [14,15,16]. Inverted PVSCs having the planar structure feature the perovskite layer coated on a HTL, with the ETL presented on the perovskite layer [17,18,19,20,21]. Such PVSCs are attractive because they ensure simple and cost-effective solar cell fabrication. Moreover, inverted PVSCs employing PC_61_BM as the ETL have exhibited relatively small hysteresis while maintaining excellent PCEs and stabilities [22].

The polymeric hole transporting materials (HTM) that have typically been used to form the HTLs of inverted PVSCs include poly(3,4-ethylenedioxythiophene):polystyrene sulfonate (PEDOT:PSS) [23,24,25], poly[bis(4-phenyl)(2,4,6-trimethylphenyl)amine] (PTAA) [26,27,28], and P3CT [29,30,31]. Nevertheless, polymeric HTM-based PVSCs have tended to exhibit limited open-circuit voltages (*V*_OC_) and lower PCEs, due to a mismatch in the work functions of the HTL and perovskite layer [32,33]. The high acidity and hygroscopicity of a polymeric HTL usually lead to unstable performance for inverted PVSCs [32,33,34]. Accordingly, inorganic nickel oxide (NiO*_x_*) materials have become more attractive for use as HTLs in inverted PVSC because of their low cost, ease of synthesis, high optical transmission, wide band gap, high hole mobility, deep valence band that matches well with those of perovskites, and excellent thermal and chemical stabilities [35,36,37,38,39,40,41]. Any enrichment of Ni^3+^ ions can, however, result in poor electrical conductivity of the NiO*_x_* layer, leading to the accumulation of holes at the NiO*_x_*–perovskite interface and, thereby, undesired charge recombination and inefficient collection of charge [42,43]. Moreover, a higher energy barrier at the NiO*_x_*–perovskite interface would inhibit hole transfer from the perovskite layer to the NiO*_x_* layer, resulting in hole accumulation [44]. Apart from that, poor charge extraction of NiO*_x_* can lead to hysteresis of the photocurrent density–voltage characteristics [43]. Poor contact at the interface between the NiO*_x_* and perovskite layers affects not only the interfacial charge transfer, but also the growth of perovskite crystallites [45]. The presence of these interfacial NiO*_x_* defects can decrease the photovoltaic (PV) performance of inverted PVSCs [46].

At present, the primary strategy for repairing contact defects at the NiO*_x_*–perovskite interface is to modify the surface morphology, physical properties, and electrical properties of the NiO*_x_* film. Many kinds of interfacial modifiers have been inserted between the NiO*_x_* and perovskite layers, including alkali metal halides [47,48], inorganic nanoparticles [36,49], nano-carbon materials [43], Lewis bases [50,51,52,53,54], Lewis acids [55], organic dyes [56,57,58,59,60], and polymers [61,62]. Inserting a thin interlayer between the NiO*_x_* and perovskite layers can repair the surface defects of the NiO*_x_*, enhance the surface wettability, minimize the energy offset between the NiO*_x_* and the perovskite, and improve the crystal growth of the perovskite [46,63,64]. Poly(vinyl butyral) (PVB) has been used as a template to produce NiO*_x_* layers with a porous structure on compact NiO*_x_* layers [45,65]. Such bilayer NiO*_x_* films have exhibited lower surface roughness, higher surface energy, and less cracking when compared with corresponding single compact layers, leading to enhanced charge extraction, increased charge transportation, and suppressed trap-assisted recombination at the NiO*_x_*–perovskite interface [65]. Moreover, MAPbI_3_-based perovskite films deposited on these bilayer-NiO*_x_* films have exhibited improved surface coverage, enlarged grains, and higher crystallinity, thereby minimizing the number of recombination traps and improving the PV properties of the resulting PVSCs [45,65]. An interfacial layer based on a blend of ethanolamine (EA) and poly(oxyethylene tridecyl ether) (PTE), inserted at the NiO*_x_*–MAPbI_3_ interface [61], not only improved the surface morphology of the NiO*_x_* layer, but also favored the crystal growth of MAPbI_3_. As a result, the charge carrier separation and hole extraction capabilities at the interface between the NiO*_x_* and MAPbI_3_ layers were both improved, leading to greater PV performance of the EA/PTE-incorporated PVSC [61]. Wang et al. reported that a coating of polyethylenimine (PEI) cations not only passivated the surface defects of NiO*_x_* films but also induced the generation of PEI-based two-dimensional (2D) perovskite interlayers between the NiO*_x_* and MAPbI_3_ layers [62]. The presence of this PEI-based 2D perovskite interlayer mitigated lattice mismatch between the NiO*_x_* and perovskite, thereby suppressing the interfacial defects formation and ensuring high-quality crystal growth of MAPbI_3_ layer, leading to improvements in the PV performance of the PVSC.

Recently, cellulose derivatives have been incorporated into MAPbI_3_ films to improve the morphology and PV performance of PVSCs. The incorporation of ethyl cellulose (EC) results in the formation of hydrogen bonding between the EC and MAPbI_3_, thereby passivating the charge defect traps at the grain boundaries of perovskite layer [66]. The polymer chains of EC act as a scaffold for the MAPbI_3_, eliminating annealing-induced lattice strain during the fabrication process of MAPbI_3_ layer and enhancing the PV performance of EC-blended MAPbI_3_-based PVSCs. Moreover, the incorporation of EC or cellulose acetate into a MAPbI_3_ film can improve its thermal stability, moisture stability, and photostability—the result of hydrogen bonding stabilizing its crystal structure [67,68]. In addition, the transparency of hydroxyethyl cellulose (HEC) in the visible range and its interactions with perovskite precursors have led to the realization of semi-transparent MAPbI_3_–polymer composites. The addition of HEC in MAPbI_3_ improved its visible transmittance and, foremost, enhanced its thermal stability without compromising the PV properties of PVSCs [69]. More recently, we synthesized three quaternary ammonium halide-containing cellulose derivatives (PQF, PQCl, PQBr) for use as defect passivation additives in MAPbI_3_ for P3CT:Na-based inverted PVSCs [70]. The addition of PQCl in the MAPbI_3_ layer reduced the grain boundaries, enhanced the crystallinity, and increased the coverage of the MAPbI_3_ layer during the formation of the resulting MAPbI_3_ films. Nevertheless, the steric bulk of the polymer main chains of the cellulose derivatives resisted the crystal formation of MAPbI_3_, leading to a decrease in the average size of the MAPbI_3_ crystals [70]. In this present study, we used these same three cellulose derivatives (PQF, PQCl, PQBr) as interfacial modifiers between the NiO*_x_* and MAPbI_3_ layers to minimize the formation of interfacial defects and enhance the PV properties of resulting PVSCs. Figure 1 presents the chemical structures of these cellulose derivatives and the architecture of the PVSCs incorporating the cellulose derivative-modified NiO*_x_* layers. We suspected that coating with a cellulose derivative would have the potential to smoothen the surface of the NiOx layer. The presence of the hydrophilic polymer chains of cellulose would likely tune the surface wettability of the NiO*_x_* film and enhance the compatibility of the NiO*_x_* and perovskite layers. Moreover, quaternary ammonium halides have great potential for use as modifiers at NiO*_x_*–perovskite interfaces, where they can passivate interfacial defects, improve interfacial contact, and increase the crystallinity and stability of the perovskite layer [71]. In addition, the fluoride (F^−^), chloride (Cl^−^), or bromide (Br^−^) anions associated with quaternary ammonium groups are more electronegative and smaller than iodide (I^−^) anions, and they form ionic bonds with Pb^2+^ cations that are stronger than the Pb–I bonds in MAPbI_3_ films. As a result, the halide anions of the quaternary ammonium groups of cellulose derivatives have the potential to passivate uncoordinated Pb^2+^ species through Lewis acid–base interactions at interface between the NiO*_x_* and perovskite layers [72,73,74,75]. Indeed, we found that inserting a quaternary ammonium halide–containing cellulose derivative at the NiO*_x_*–MAPbI_3_ interface suppressed the formation of interfacial defects and promoted the crystal growth of the perovskite film. Therefore, the modified MAPbI_3_-based PVSCs showed improved PV properties. Herein, we describe the effects of inserting PQF-, PQCl-, and PQBr-based interfacial modifiers on the morphologies, crystal structures, and optical absorption properties of MAPbI_3_ films and on the PCEs of their PVSCs.

## 2. Experimental Details

### 2.1. Chemicals

Quaternized hydroxyethylcellulose ethoxylate (PQ-Cl; weight-average molecular weight (*M*_w_) = 250,000 g mol^−1^; degree of polymerization = 600) was purchased from Sigma–Aldrich and used as received. Bathocuproine (BCP) was purchased from Acros. 6,6-Phenyl-C_60_-butyric acid methyl ester (PC_61_BM) was purchased from Uni-onward and used as received. Methylamine (CH_3_NH_2_), lead iodide (PbI_2_), and other reagents and chemicals were purchased from Acros, (Fukuoka, Japan), Aldrich (St. Louis, MO, USA), and TCI Chemical (Tokyo, Japan) and used as received. PQF and PQBr were synthesized through ionic change of PQCl with NaF and NaBr, respectively [70].

### 2.2. Characterization of Cellulose Derivatives and Perovskite Layers

Absorption spectra of MAPbI_3_ films coated on cellulose derivative-modified NiO*_x_*-deposited fluorine-doped tin oxide (FTO) glass were recorded using a Hitachi U3010 UV–Vis spectrometer (Hitachi High-Tech Co., Tokyo, Japan). Photoluminescence (PL) spectra of the cellulose derivative–modified MAPbI_3_ films were measured using a Hitachi F-4500 fluorescence spectrophotometer (Hitachi High-Tech Co., Tokyo, Japan). Time-resolved PL (TRPL) spectra of MAPbI_3_ films coated on the PQF-, PQCl-, and PQBr-modified NiO*_x_*-deposited FTO glass were recorded using a Horiba Fluoromax-4 spectrometer and Delta Time TCSPC-MCS kit with 405-nm pulsed light emitting diode (LED). The pristine and cellulose derivative–modified MAPbI_3_ films were encapsulated for measurement of their UV–Vis, PL, and TRPL spectra. The morphologies of the cellulose derivative-modified NiO*_x_* and MAPbI_3_ layers were imaged using atomic force microscopy (AFM, Seiko SII SPA400, Chiba, Japan), performed in the tapping mode. Three runs of surface roughness measurements were performed for each MAPbI_3_ layer. The surface and cross-sectional morphologies of the MAPbI_3_ layers deposited on the cellulose derivative–modified NiO*_x_* layers were analyzed using cold field emission scanning electron microscopy (FESEM; Hitachi-4800; Integrated Service Tech. Inc., Hinchu, Taiwan; operating voltage: 1.5–2.0 kV). The crystalline structures of the MAPbI_3_ layers were determined using X-ray powder diffractometry (XRD, Shimadzu SD-D1, Shimadzu Scientific Instrument Co., Taipei, Taiwan), operated with a Cu target at 35 kV and 30 mA. The contact angles (CAs) of water droplets on the cellulose derivative-modified NiO*_x_* films were measured using a Kyowa DropMaster optical CA meter (Applied Trentech Inc., Taipei, Taiwan).

### 2.3. Fabrication and Characterization of PVSCs

The PVSCs in this study had the structure FTO-deposited glass/NiO*_x_*/cellulose derivative/MAPbI_3_/PC_61_BM/BCP/Ag (100 nm), where the NiO*_x_* layer was modified with a quaternary ammonium halide-containing cellulose derivative (PQF, PQCl, or PQBr). FTO-deposited glass (sheet resistance: 7 Ω square^−1^) was purchased from Solaronix. The FTO substrates of PVSCs with patterned electrodes were washed well and then cleaned through O_2_ plasma treatment. The NiO*_x_* precursor solution was prepared by dissolving nickel(II) acetate tetrahydrate (100 mg) in isopropanol and ethanolamine, stirring at 70 °C for several hours, and then filtering through a 0.45-μm polytetrafluoroethylene (PTFE) based filter. The NiO*_x_*-based HTL was deposited on the FTO layer through spin-coating of the NiO*_x_* precursor solution [50]. The sample was dried at 80 °C for 10 min and then thermally treated at 450 °C for 60 min. Various amounts of PQX cellulose derivative (X = F, Cl, Br; 0.03, 0.05, or 0.1 wt.%) were dissolved in DI water. The resulting solution was deposited on the surface of the NiO*_x_*-based HTL. The sample was dried at 100 °C for 30 min. The NiO*_x_* layers modified with 0.03, 0.05, and 0.10 wt.% of PQF are named herein as PQF-0.03, PQF-0.05, and PQF-0.10, respectively; the films modified with 0.03, 0.05, and 0.10 wt.% of PQCl are named PQCl-0.03, PQCl-0.05, and PQCl-0.10, respectively; and the films modified with 0.03, 0.05, and 0.10 wt.% of PQBr are named PQBr-0.03, PQBr-0.05, and PQBr-0.10, respectively. The MAI and PbI_2_ were stirred in a mixture of DMF and DMSO (4:1, *v*/*v*). The MAI and PbI_2_ containing solution was deposited on top of the cellulose derivative–modified NiO*_x_*-based HTL. The MAPbI_3_ deposited substrate was dried at 100 °C for 10 min. Next, a solution of PC_61_BM in CB (20 mg mL^−1^) was deposited on top of the MAPbI_3_ layer. A solution (0.3 mL) of BCP in isopropanol (0.5 mg mL^−1^) was then deposited on the PC_61_BM layer. The Ag-based cathode was thermally deposited onto the PC_61_BM layer in a high-vacuum chamber. The photo-active area of the cell was 0.20 cm^2^. The PV properties of the PVSCs were measured using a programmable electrometer equipped with current and voltage sources (Keithley 2400) under illumination with solar-simulating light (100 mW cm^−2^) from an AM1.5 solar simulator (NewPort Oriel 96000).

## 3. Results and Discussion

### 3.1. Characterization of Cellulose Derivative–Modified NiO_x_ Layers

Because the PV parameters of PVSCs is closely related to the morphology of their NiO*_x_*-based HTLs, we used SEM and AFM to investigate the morphologies of the cellulose derivative (PQF, PQCl, PQBr)–deposited NiO*_x_* layers. SEM images of the PQCl-coated NiO*_x_* layer on the FTO substrate are shown in Figure 2. In Figure 2a, we observe the nanosheet structure of the pristine NiO*_x_*. The shape and size of the NiO*_x_* nanosheets did not change significantly after coating with different concentrations of PQCl (Figure 2b–d), suggesting the presence of thin films of this cellulose derivative. The surface morphologies of the PQF- and PQBr-deposited NiO*_x_* layers were similar to those of the PQCl-deposited NiO*_x_* layers (Appendix A). Figure 3 presents topographic images of the PQCl-coated NiO*_x_* layers on the FTO substrates. These AFM images indicate that the surface morphology of the NiO*_x_* layer changed after coating with PQCl, with a nanoparticle structure appearing. Nevertheless, the surface morphology was not changed significantly when coating with different concentrations of PQCl. The surface roughness of the PQCl-modified NiO*_x_* layers was slightly enhanced when compared with that of the pristine NiO*_x_* layer (Table 1). We observed similar features in the AFM images of the PQF- and PQBr-deposited NiO*_x_* layers (Appendix A). Furthermore, we used a CA meter to examine the hydrophobicity/hydrophilicity of the surface-modified NiO*_x_* layers. Appendix A displays photographs of water droplets on the pristine and cellulose derivative-deposited NiO*_x_* layers. Table 1 reveals that the CAs of the cellulose derivative-modified NiO*_x_* layers were lower than that of the pristine NiO*_x_* layer. Shen et al. reported that enhancing the hydrophilicity of NiO*_x_*-based HTLs encourages the formation of more uniform and larger crystal grains in MAPbI_3_ layers [45]. Nevertheless, the uniformity and crystal size of MAPbI_3_ were not only affected by the wettability of the HTL.

### 3.2. Morphologies of Perovskite Films Deposited on Cellulose Derivative–Modified NiO_x_ Layers

To investigate the effects of the quaternary ammonium halide-functionalized cellulose derivatives (PQF, PQCl, PQBr) as interfacial layers on the crystallization of the perovskite films, we used SEM to examine the morphologies and film qualities of MAPbI_3_ deposited on the cellulose derivative–modified NiO*_x_* layers, thereby allowing us to determine the optimal processing conditions for the preparation of the PVSCs. Figure 4 and Appendix A display the top-view and cross-sectional SEM images, respectively, of MAPbI_3_ films that had been deposited on the interfacial modifiers PQCl, PQF, and PQBr that had been subjected to annealing at 100 °C for 10 min. Moreover, Figure 5, Appendix A and Appendix A present the crystal grain size distributions of the MAPbI_3_ films deposited on the PQCl, PQF, and PQBr interfacial layers, respectively. Table 1 summarizes the average crystal sizes of the PQCl-, PQF-, and PQBr-based MAPbI_3_ films, calculated using Image J1 software. The crystal grains that appeared after growing the MAPbI_3_ layer on the PQCl-modified NiO*_x_* HTL were larger than those of the pristine NiO*_x_* HTL. The largest crystal grains of MAPbI_3_ were those for the sample prepared using the PQCl-0.05–modified NiO*_x_*. Nevertheless, the standard derivation (SD) of the crystal grain size distribution of the PQCl-0.05–modified MAPbI_3_ layer was slightly larger than that of the pristine sample. Figure 6 provides a schematic representation of the crystal growth of an MAPbI_3_ film on the PQCl-modified NiO*_x_*-based HTL. The quaternary ammonium halide units of PQCl have a chemical structure similar to that of MAI, suggesting that they might participate in the perovskite crystallization process through partial substitution of the MA cations with the quaternary ammonium cations as well as of the I^−^ anions with Cl^−^ anions [76,77]. The quaternary ammonium halide-containing side chains of the cellulose derivative PQCl-0.05 on the surface of the NiO*_x_* layer appeared to help with the repair of the crystal defects and promoted the crystal growth of MAPbI_3_, encouraging the formation of more uniform and larger crystal grains in the perovskite film [76]. When we coated a higher content of the cellulose derivative (PQCl-0.10) on the surface of the NiO*_x_* layer, interfusion of the large polymer backbone into the perovskite layer occurred during crystal formation in the MAPbI_3_ layer, thereby decreasing the average size of the perovskite crystals (Figure 6) [67,76]. The corresponding effects of PQF and PQBr at repairing the crystal defects were much poorer than that of PQCl [67,77]. Moreover, the average sizes of the MAPbI_3_ crystal grains coated on the PQF- and PQBr-deposited NiO*_x_* were smaller than that of the pristine NiO*_x_*. The average sizes of the MAPbI_3_ crystal grains decreased upon coating the cellulose derivatives PQF and PQBr at higher concentrations onto the surface of NiO*_x_* layer. Cross-sectional SEM images indicated that the crystal grains of MAPbI_3_ became more densely packed after inserting an interfacial layer of PQCl between the NiO*_x_* and MAPbI_3_ layers. Relative to the pristine MAPbI_3_ film, the grain boundaries between the various crystal grains became vaguer for the cellulose derivative-incorporated MAPbI_3_ films, resulting in higher coverage of the perovskite films [67]. The repairing of crystal defects mediated by the quaternary ammonium halides presumably helped to modify the grain boundaries [77]. The minimization of grain boundaries and the enhanced packing density of crystal grains would presumably be favorable for charge transfer in the perovskite films. The cross-sectional SEM images indicated that the thickness of the perovskite layer did not change significantly after increasing the PQCl or PQF content (Figure 4 and Appendix A), but it did decrease for the PQBr-modified perovskite layer. A thinner MAPbI_3_ layer would presumably result in a lower capacity to absorb solar light and poorer PV performance from the corresponding PVSC.

AFM microscopy confirmed the interfacial effects of the PQF, PQCl, and PQBr on the morphologies of the MAPbI_3_ films. Figure 7, Appendix A and Appendix A present AFM images of the MAPbI_3_ films deposited on the interfacial layers of PQCl, PQF, and PQBr, respectively. Table 1 summarizes the statistical surface roughness of the MAPbI_3_ films deposited on the PQCl, PQF, and PQBr interfacial layers. The AFM images indicate that largest crystal grains appeared after growing the MAPbI_3_ layer on the PQCl-modified NiO*_x_* HTL. Moreover, the average size of the MAPbI_3_ crystal grains decreased when coating the NiO*_x_* layer with a higher concentration of PQCl. The surface roughness of the MAPbI_3_ films coated on the cellulose-modified NiO*_x_* layers was slightly lower than that on the pristine NiO*_x_* layer. The surface roughness of the MAPbI_3_ films was slightly higher when the NiO*_x_* film had been coated using a solution of 0.10 wt.% of the cellulose derivative, relative to those obtained using the 0.03 and 0.05 wt.% solutions. Inserting the PQCl at the NiO*_x_*–MAPbI_3_ interface promoted the formation of more uniform and larger crystal grains, and decreased the surface roughness of the MAPbI_3_ film, presumably through the defect passivation effect of PQCl [78]. We suspected that a lower degree of light scattering and a higher absorption capacity, both favorable for enhancing PV properties, would be obtained for MAPbI_3_ films having smoother surfaces and better film quality [79,80]. Nevertheless, the modification effects of PQF and PQBr at the MAPbI_3_–NiO*_x_* interfaces were much poorer than that of PQCl. The average sizes of the MAPbI_3_ crystal grains coated on the PQF- and PQBr-deposited NiO*_x_* were smaller than that on the pristine NiO*_x_*.

### 3.3. XRD Images of Perovskite Films Deposited on Cellulose Derivative–Modified NiO_x_ Layers

XRD was used to examine the crystal structures of the MAPbI_3_ films deposited on the cellulose derivative-modified NiO*_x_* layers. Figure 8, Appendix A and Appendix A reveal that the patterns of the MAPbI_3_ films formed on the PQF-, PQCl-, and PQBr-modified NiO*_x_* layers featured the typical diffraction peaks of MAPbI_3_ based perovskites, including characteristic peaks at 14.2, 28.4, and 43.08° corresponding to the (110), (220), and (330) phases, respectively [81,82,83]. These diffraction peaks indicated the formation of tetragonal crystal structures having lattice constants *a* and *b* each equal to 8.883 Å and *c* equal to 12.677 Å [82]. Moreover, the intensities of the (110) peaks for the MAPbI_3_ films coated on the PQCl-0.03- and PQCl-0.05-modified NiO*_x_* layers were higher than that for the MAPbI_3_ coated on the pristine NiO*_x_* layer (Figure 8). The highest intensity of the (110) peak was that for the MAPbI_3_ film deposited on the PQCl-0.05–modified NiO*_x_* layer. A higher (110) diffraction peak intensity correlates with a better crystal quality for MAPbI_3_ films [45,51,65,70]. An MAPbI_3_ film of better crystal quality tends to display improved electronic properties, including greater charge carrier transport [45,51,65,70]. The presence of quaternary ammonium cations and Cl^−^ anions at the NiO*_x_*–MAPbI_3_ interface can passivate the positively charged defects in the perovskite layer induced by the loss of I^−^ anions. Furthermore, the ammonium unit can passivate Pb–I antisite defects through electrostatic interactions [76]. Therefore, we found that the crystal growth of MAPbI_3_ was promoted through the crystal defect repairing effect of PQCl. In contrast, the diffraction intensities of the (110) peaks for the MAPbI_3_ layers deposited on the PQF- and PQBr-modified NiO*_x_* layers were lower when compared with that of the MAPbI_3_ deposited on the pristine NiO*_x_* layer (Appendix A). Relative to the effect of PQCl, the interfacial layers of PQF and PQBr led to poorer crystal growth of the perovskite. We suspect that greater electronegativity limited the dissociation of F^−^ anions from the quaternary ammonium fluoride, such that fewer F^−^ anions could compensate for the I^−^ vacancies of the perovskite [79,84]. Furthermore, the relatively large ionic radius of the Br^−^ anion would affect its ability to compensate for ion of I^−^ vacancies. As a result, the crystal defect repairing effects of PQF and PQBr were both poorer than that of PQCl [70]. Table 1 summarizes the crystal sizes in the MAPbI_3_ films coated on the PQF-, PQCl-, and PQBr-modified NiO*_x_* layers. According to the Scherrer equation, these crystal sizes were calculated from the full width at half maximum (FWHM) of the (110) diffraction peak [85]. The average crystal sizes were greatest for the MAPbI_3_ layers that had been deposited on the PQCl-modified NiO*_x_* layers. The largest crystals were those in the PQCl-0.05-based MAPbI_3_ film. The crystal sizes were lower for the MAPbI_3_ layers deposited on the PQF- and PQBr-modified NiO*_x_* layers, and they decreased for the NiO*_x_* layers that had been treated with higher concentrations of the PQF and PQBr solutions. We attribute the smaller crystals to the presence of a higher content of polymer chains at the MAPbI_3_–NiO*_x_* interface. The steric bulk of the cellulose derivative-based polymer backbone presumably inhibited the formation of crystals of MAPbI_3_, leading to smaller crystals of the MAPbI_3_ [70].

### 3.4. UV–Vis Absorption Spectra of MAPbI_3_ Films Deposited on Cellulose Derivative–Modified NiO_x_ Layers

We recorded UV–Vis spectra of the MAPbI_3_ films that had been deposited on the cellulose derivative–modified NiO*_x_* layers to examine the effects of the interfacial layers on the optical absorption properties of the perovskite films (Figure 9). Compared with the pristine MAPbI_3_ film, the MAPbI_3_ films deposited on the PQCl-0.03- and PQCl-0.05-modified NiO*_x_* layers absorbed more strongly over almost the entire spectral range. The highest absorption intensity was that for the PQCl-0.05-based MAPbI_3_ film, consistent with its greater crystallinity, minimized grain boundaries, increased coverage, and lower reflection. Notably, however, the absorption intensity of the PQCl-0.10-based MAPbI_3_ film was lower than that of the pristine MAPbI_3_ film. Furthermore, as compared with the pristine MAPbI_3_ film, the absorption intensities were lower for the MAPbI_3_ films deposited on the PQF- and PQBr-modified NiO*_x_* layers.

PL spectroscopy was used to study the interfacial effects of the cellulose derivatives PQF, PQCl, and PQBr on the PL properties of the MAPbI_3_ films. Figure 10a displays the PL spectra of the MAPbI_3_ films deposited on the cellulose derivative-modified NiO*_x_* layers. The wavelength of maximal PL of the MAPbI_3_ films appeared near 768 nm. Relative to the signal for the MAPbI_3_ film coated on the pristine NiO*_x_* layer, the PL intensities were lower for the MAPbI_3_ films deposited on the PQCl-0.03- and PQCl-0.05-modified NiO*_x_* layers, implying that the charge separation capacity was enhanced for the MAPbI_3_ perovskite films deposited on the PQCl interfacial layers [24]. Moreover, the PL intensity of the MAPbI_3_ film deposited on PQCl-0.05 was lower than those of the PQCl-0.03- and PQCl-0.10-based MAPbI_3_ films. We attribute the low PL intensity of the PQCl-0.05-modified MAPbI_3_ film to the decrease in the number of crystal defects and the excellent defect passivation occurring at the NiO*_x_*–MAPbI_3_ interface. Relative to the signal for the pristine MAPbI_3_ film, the PL intensities were enhanced for the MAPbI_3_ layers deposited on the PQF- and PQBr-modified NiO*_x_* layers, implying that PQF and PQBr could not passivate the interfacial defects and, thereby, inferior interfacial contact and poorer charge-separation capacity occurred at their NiO*_x_*–MAPbI_3_ interfaces.

TRPL spectra were used to study the influence of the interfacial modifiers PQF, PQCl, and PQBr on the charge recombination processes of the perovskite films (Figure 10b). The carrier lifetime was obtained by fitting the PL data to a double-exponential decay model [86,87]:I(*t*) = *Ae*^−t/τ1^ + *Be*^−t/τ2^
where *A* and *B* are constants and τ_1_ and τ_2_ are the fast and slow decay constants, respectively. The fitting results for the TRPL spectra are summarized in Table 2. Here, the average lifetime of the cell was calculated from the average of the fast and slow decay constants, obtained using the equation
τ_avg_ = (*A*τ_1_^2^ +*B*τ_2_^2^)/(*A*τ_1_ + *B*τ_2_)

The constant τ_1_ is related to defect recombination or interfacial charge transport from MAPbI_3_ to the HTLs; τ_2_ is related to radiative recombination [86]. The lifetimes for the PQCl-0.03- and PQCl-0.05-modified MAPbI_3_ films samples were shorter than that for the pristine MAPbI_3_ film, indicating that the addition of PQCl as an interfacial modifier could minimize the number of defects in the MAPbI_3_ film, enhance the degree of charge extraction, and decrease the non-radiative combination loss. Nevertheless, inserting an excess of PQCl at the NiO*_x_*–MAPbI_3_ interface did not lower the number of defects of the MAPbI_3_ film, with the PQCl-0.10 sample exhibiting a longer lifetime than that of the pristine MAPbI_3_ film. Apart from that, the lifetimes of the PQF- and PQBr-modified MAPbI_3_ films were longer than that of the pristine MAPbI_3_ layer, and they increased significantly when higher contents of PQF and PQBr were present at the NiO*_x_*–MAPbI_3_ interfaces. The TRPL spectra indicated that the carrier lifetimes of the PQCl-modified MAPbI_3_ films were much shorter than those of the MAPbI_3_ films coated on the PQF and PQBr layers.

### 3.5. PV Properties of PVSCs Incorporating Cellulose Derivative–Modified NiO_x_ Layers

The optimized spin-coating procedure was used to prepare PVSCs incorporating the cellulose derivatives at the NiOx–MAPbI_3_ interfaces. Figure 11 presents the photocurrent density–voltage plots of the PVSCs fabricated using the NiO*_x_* layers modified with various contents of the cellulose derivatives PQF, PQCl, and PQBr. Figure 12 displays statistical box plots for the PV parameters of 20 un-encapsulated pristine and cellulose derivative–modified MAPbI_3_-based PVSCs. The statistical values of the PV properties of these PVSCs are summarized in Table 3, including their open-circuit voltages (*V*_OC_), short-circuit current densities (*J*_SC_), fill factors (FFs), and PCEs. We performed 20 runs of PV evaluation measurements for each cell. A value of *V*_OC_ of 1.07 V, a value of *J*_SC_ of 17.98 mA cm^−2^, an FF of 69.3%, and a PCE of 13.33% were obtained for PVSC I, fabricated from the NiO*_x_* HTL prepared without modification with a cellulose derivative as interfacial layer. These PV parameters are comparable with those of other published PVSC having similar architectures [87]. Relative to the pristine PVSC I, we obtained superior PV properties for PVSCs V and VI, based on the PQCl-0.03– and PQCl-0.05–modified NiO*_x_* HTLs, respectively, but not for PVSC VII. The PV properties enhanced after increasing the content of PQCl at the MAPbI_3_–NiO*_x_* interfaces for PVSCs V and VI. The highest performance was that for PVSC VI, incorporating the PQCl-0.05 film: a value of *V*_OC_ of 1.06 V, a value of *J*_SC_ of 18.35 mA cm^−2^, an FF of 74.0%, and a PCE of 14.40%. These high values of *V*_OC_, *J*_SC_, and FF are consistent with the higher crystallinity and stronger UV–Vis absorptions of the MAPbI_3_ films deposited on the PQCl-modified NiO*_x_* layers. The presence of PQCl at the MAPbI_3_–NiO*_x_* interface, with Cl^−^ anions in the quaternary ammonium units, promoted the crystal growth and enhanced the MAPbI_3_ film quality, leading to efficient charge separation and extraction and a low degree of charge recombination [88]. Incident photon-to-current efficiency (IPCE) spectroscopy confirmed the improvements in the values of *J*_SC_ of the PVSCs fabricated from the cellulose derivative–modified NiO*_x_* HTLs (Figure 13). The IPCEs of the PQCl-0.03- and PQCl-0.05-based PVSCs V and VI, respectively, were higher than that of PVSC I fabricating from the pristine NiOx film. Thus, we conclude that the incorporation of PQCl at the MAPbI_3_–NiO*_x_* interface had significant effects on repairing the crystal defects and enhancing the crystallinity of the MAPbI_3_ film. As a result, the PV properties were improved for the PVSCs fabricated from the PQCl-0.03 and PQCl-0.05 samples. In addition, the PV properties of the PQCl-0.10-based PVSC IV were poorer than those of PVSC I (based on the pristine NiO*_x_* HTL), consistent with the lower crystallinity (as evidenced from SEM and AFM images and XRD patterns) of the PQCl-0.10–modified MAPbI_3_ film. Nevertheless, the presence of an excessive amount of PQCl at the MAPbI_3_–NiO*_x_* interface did not improve the PV performance. When a higher content of PQCl was coated on the surface of the NiO*_x_* layer, the sterically bulky polymer interfused among the MAPbI_3_ crystals and limited their growth, thereby decreasing the crystallinity, the absorption intensity, and the PV performance of the PQCl-0.10-based PVSC IV. In addition, the PV properties of the PVSCs were poorer when the MAPbI_3_ films were deposited on the PQF- and PQBr-modified NiO*_x_* HTLs, implying that the effects of PQF and PQBr on crystal defect repair were much poorer than that of PQCl. Moreover, the PV performance of the PVSCs decreased when higher amounts of PQF (PVSCs II, III, and IV) and PQBr (PVSCs VIII, IX, and X) were present at the MAPbI_3_–NiO*_x_* interfaces. Nevertheless, the PCEs of the PQF-modified PVSCs (PVSCs II, III, and IV) were slightly higher than those of the PQBr-modified PVSCs (PVSCs VIII, IX, and X). Compared with the PQBr-modified MAPbI_3_ layers, the higher UV–Vis spectral absorption intensities and larger average crystal grain sizes of the PQF-modified MAPbI_3_ films resulted in the higher IPCEs and PV performance parameters of PVSCs II–IV. Based on these findings, we conclude that the PV performance of MAPbI_3_-based PVSCs can be improved through modification of the MAPbI_3_–NiO*_x_* interface with an optimized amount of PQCl, which has a positive effect on crystal growth and crystal defect repair in the MAPbI_3_ layer.

We measured the hole mobility in the MAPbI_3_ layers to further examine the passivation effects of the cellulose derivative-based interfacial modifiers on the perovskite layers (Figure 14). We calculated the mobility (*μ*) of the perovskite in the space-charge limited current regime using the equation
*J* = 9/8 *ε*_r_*ε*_o_*μV*^2^/*L*^3^
where *J* is the current density, *ε*_o_ is the vacuum permittivity (8.854 × 10^−12^ F m^−1^), *ε*_r_ is the relative permittivity of MAPbI_3_ (32), *V* is the base voltage, and *L* is the thickness of the MAPbI_3_ layer (410 nm) [66,89,90]. The estimated hole mobilities of the pristine and PQF-0.05-, PQCl-0.05-, and PQBr-0.05-based hole-only devices were 3.92 × 10^−3^, 2.68 × 10^−3^, 4.20 × 10^−3^, and 2.14 × 10^−3^ cm^2^ V^−1^ s^−1^, respectively. Thus, the hole mobility of the PQCl-0.05–modified MAPbI_3_ layer was greater than that of the pristine sample, while those of the PQF-0.05– and PQBr-0.05–modified samples were lower. We infer that the passivation effect of PQCl on the perovskite layer was much better than those of PQF and PQBr.

To further examine the effects of the addition of PQCl on the morphologies and optical properties of the MAPbI_3_-based perovskite films, we prepared a perovskite film (MAPbI_3_:PQCl-0.06) from a blend of PQCl (0.06 wt.%) and MAPbI_3_ deposited on the surface of the PQCl-0.05–modified NiO*_x_* layer. The SEM images in Figure 15a,b reveal that the average crystal grain size (111 nm) of the MAPbI_3_:PQCl-0.06 film was much lower than that (272 nm) of the pristine MAPbI_3_ film (Figure 4c). We suspect that the steric bulk and low thermal mobility of the large cellulose derivative backbones suppressed the formation of crystals of MAPbI_3_, thereby decreasing their average size [70]. Nevertheless, these crystal grains of smaller size underwent denser packing. As compared with the pristine MAPbI_3_ film, the grain boundaries among the various crystal grains were more vague for the MAPbI_3_:PQCl-0.06 film, resulting in a higher coverage of the perovskite film. The repairing of crystal defects induced by the quaternary ammonium halide units presumably helped to connect the crystal grains [76]. The decrease in the number of grain boundaries and the greater packing density of the crystal grains would both favor charge transfer in the perovskite film [78]. The XRD patterns for the perovskite films coated on the PQCl-0.05-modified NiO*_x_* layer indicated (Figure 15c) that the crystal diffraction intensity of the MAPbI_3_:PQCl-0.06 blend film was slightly lower than that of MAPbI_3_ film. Moreover, the diffraction intensity of the MAPbI_3_:PQCl-0.06 blend film was greater than that of the MAPbI_3_ film coated on the pristine NiO*_x_* layer. In addition, the PL intensity of the MAPbI_3_:PQCl-0.06 blend film was lower than that of the MAPbI_3_ film, implying that the charge separation capacity was enhanced after the addition of PQCl in the MAPbI_3_ perovskite layer [24]. Consequently, the PVSC XI device (FTO/NiO*_x_*/PQCl-0.05/MAPbI_3_:PQCl-0.06/PC_61_BM/BCP/Ag) fabricated from the MAPbI_3_:PQCl-0.06 blend film exhibited values of *J*_SC_ and PCE higher than those of the MAPbI_3_-based PVSCs I and VI (Figure 11). Indeed, the PVSC XI exhibited a PCE of 16.53%, a value of *V*_OC_ of 1.06 V, a value of *J*_SC_ of 21.93 mA cm^−2^, and an FF of 71.0% (Table 3). A high efficiency of 16.53% from forward scanning and a comparable efficiency of 16.48% from reverse scanning were obtained for the PVSC XI (Appendix A). A negligible hysteresis of the current density–voltage curve implies the balanced charge transport at the NiO_X_/MAPbI_3_ interface and good charge transport inside the MAPbI_3_ layer for the PVSC XI [19]. Furthermore, the IPCE of PVSC XI incorporating the MAPbI_3_:PQCl-0.06 blend film was higher than those of the MAPbI_3_-based PVSCs I and VI (Figure 13).

The storage stability of the cellulose derivative-based PVSCs (PVSC I, PVSC II, PVSC VI, and PVSC-XI) measured at 30 °C and 60% relative humidity is displayed in Figure 16. The PCE-stability of the PQCl incorporated PVSC VI and PVSC XI was superior to those of the Pristine, PQF-0.03-, and PQBr-0.03-based PVSCs (PVSC-I, PVSC-II, and PVSC-VIII). The lifetime of PQCl-based PVSC VI and PVSC XI without encapsulation was more than 900 h. The incorporation of PQCl at MAPbI_3_/NiO_X_ interface promotes the crystal growth and effective crystal defect passivation for stabilizing perovskite crystal structures. Moreover, the stability of the PVSC was further enhanced by the addition of PQCl-0.06 in the MAPbI_3_ layer for the PVSC XI. The interfacial layer effect of PQCl on the PV stability was much better than those of PQF and PQBr.

## 4. Conclusions

We used a series of cellulose derivatives (PQF, PQCl, PQBr) individually as interfacial modifiers of MAPbI_3_–NiO*_x_* interfaces and prepared corresponding PVSCs. The presence of quaternary ammonium cations and Cl^−^ anions at the NiO*_x_*–MAPbI_3_ interface can passivate the positively charged defects in the perovskite layer induced by the loss of I^−^ anions. Moreover, the ammonium unit can passivate Pb–I antisite defects through electrostatic interactions. The deposition of an appropriate amount of PQCl on the NiO*_x_* layer led to repair of the grain boundary defects, promoted crystal growth, and increased the light absorption and hole mobility of the MAPbI_3_ film. Nevertheless, the deposition of an excess of POCl on the NiO*_x_* suppressed crystal growth of the perovskite through the effect of the steric bulk of the polymer backbone of PQCl. Relative to the effect of PQCl, the interfacial layers of PQF and PQBr led to poorer crystal growth of the perovskite. The PV properties of PVSCs fabricated with PQCl-modified NiO*_x_* layers were improved when compared with those of the pristine sample. Furthermore, the PV parameters of a PQCl-modified, NiO*_x_*-based PVSC were further enhanced after blending the MAPbI_3_ with PQCl. As compared with the pristine MAPbI_3_ film, the grain boundaries among the various crystal grains became more vague in the MAPbI_3_:PQCl-0.06 film, resulting in a higher coverage of the perovskite film. The decrease in the number of grain boundaries and the greater packing density of the crystal grains both promoted charge transfer in the MAPbI_3_ film.

## Figures and Tables

**Figure 1 polymers-15-00437-f001:**
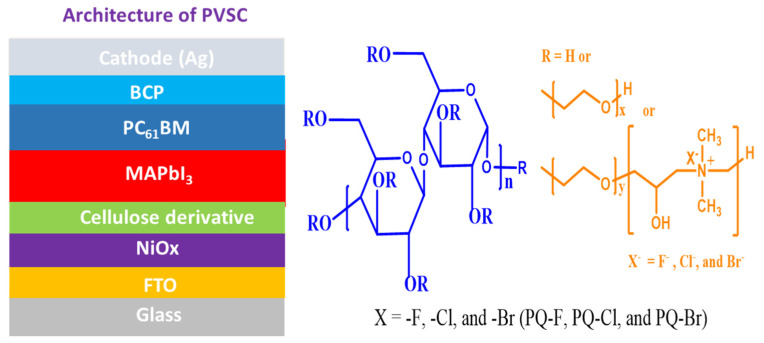
Chemical structures of the cellulose derivatives PQF, PQCl, and PQBr and architecture of the PVSCs incorporating the cellulose derivative–modified NiO*_x_* films.

**Figure 2 polymers-15-00437-f002:**
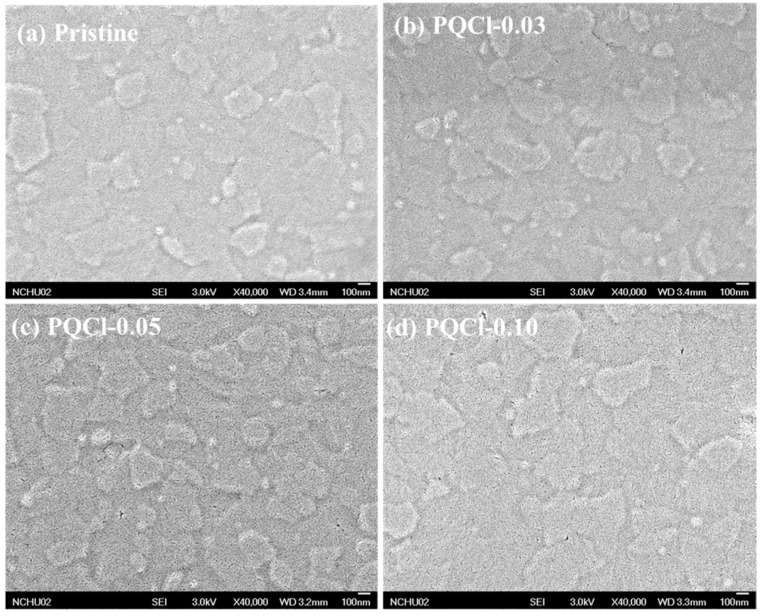
SEM images of the PQCl-deposited NiO*_x_* layers: (**a**) pristine, (**b**) PQCl-0.03, (**c**) PQCl-0.05, and (**d**) PQCl-0.10.

**Figure 3 polymers-15-00437-f003:**
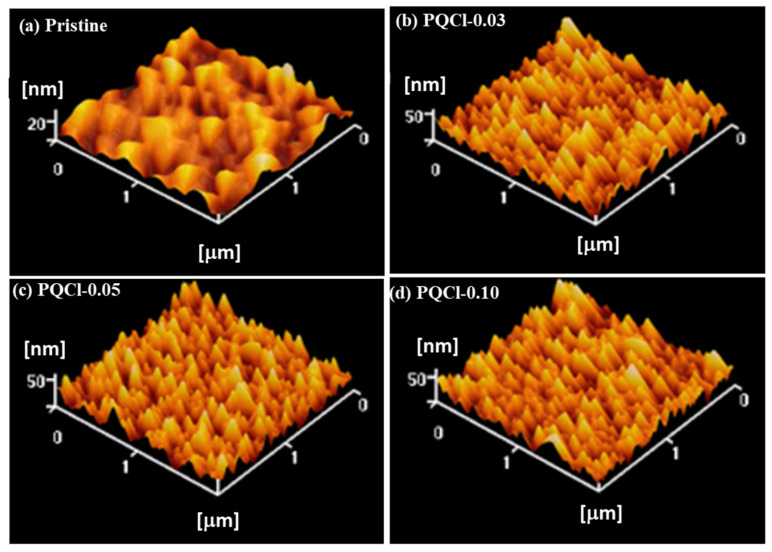
Topographic images of PQCl-deposited NiO*_x_* layers: (**a**) pristine, (**b**) PQCl-0.03, (**c**) PQCl-0.05, and (**d**) PQCl-0.10.

**Figure 4 polymers-15-00437-f004:**
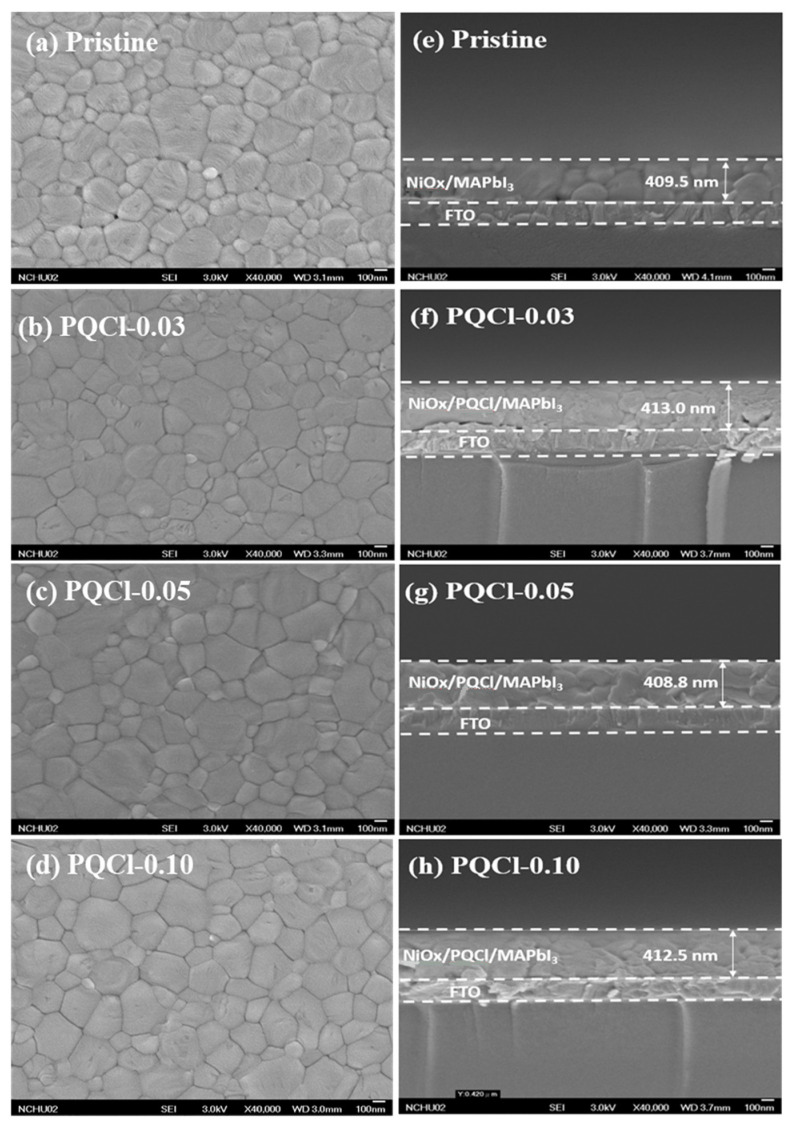
(**a**–**d**) Top-view and (**e**–**h**) cross-sectional SEM images of (**a**,**e**) pristine and, (**b**,**f**) PQCl-0.03–, (**c**,**g**) PQCl-0.05–, and (**d**,**h**) PQCl-0.10–based perovskite films.

**Figure 5 polymers-15-00437-f005:**
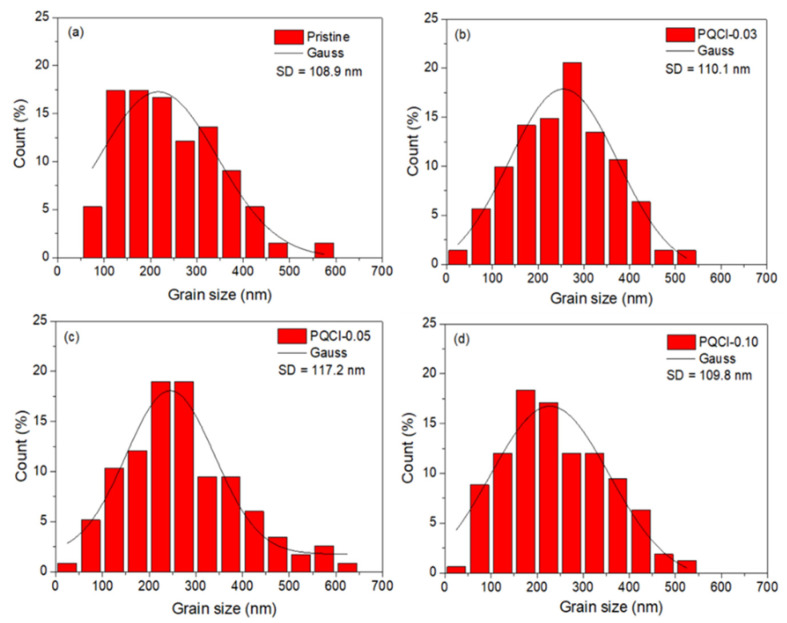
Grain size statistics and Gauss fits of the crystal grain size distributions of (**a**) pristine, (**b**) PQCl-0.03–, (**c**) PQCl-0.05–, and (**d**) PQCl-0.10–based perovskite films (SD: standard derivation).

**Figure 6 polymers-15-00437-f006:**
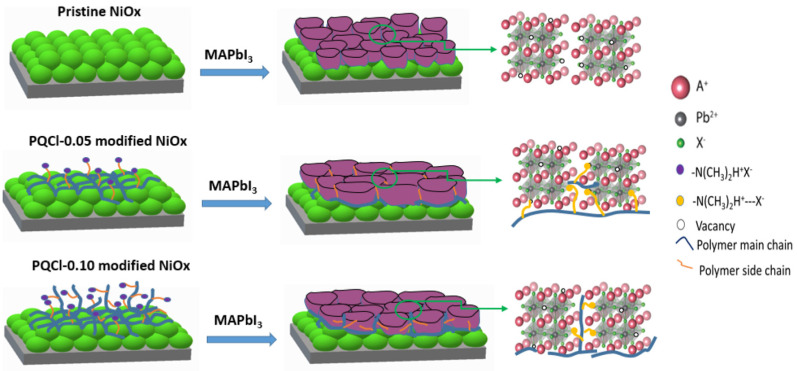
Schematic representation of crystal growth of MAPbI_3_ films on cellulose derivative PQCl–modified NiO*_x_*-based HTLs.

**Figure 7 polymers-15-00437-f007:**
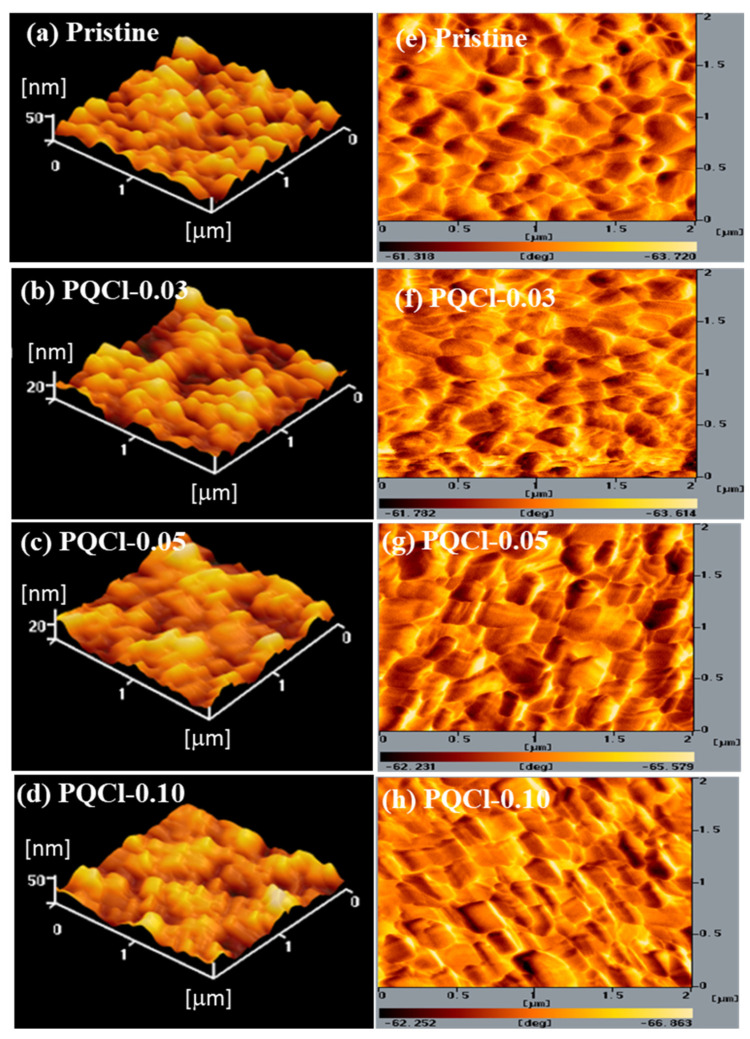
(**a**–**d**) Topographic and (**e**–**h**) phase AFM images of perovskite films deposited on (**a**,**e**) pristine and (**b**,**f**) PQCl-0.03–, (**c**,**g**) PQCl-0.05–, and (**d**,**h**) PQCl-0.10–modified NiO.

**Figure 8 polymers-15-00437-f008:**
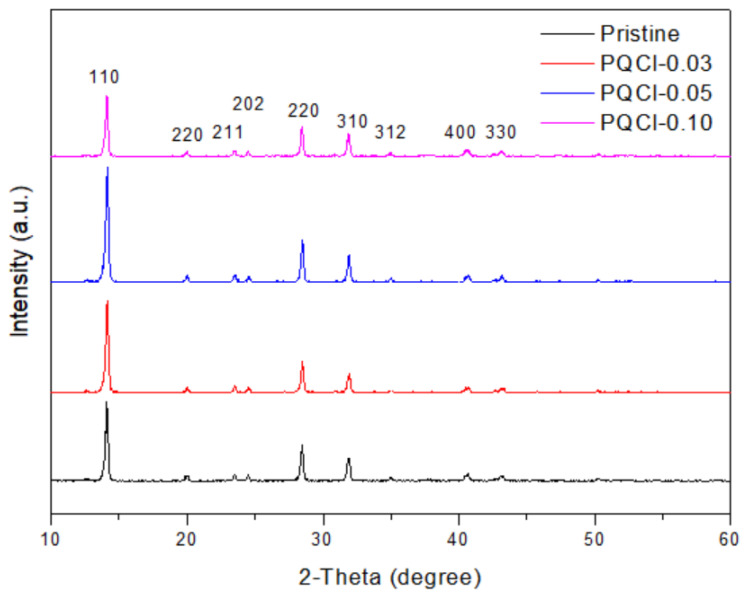
XRD patterns of MAPbI_3_ films deposited on PQCl-modified NiO*_x_* layer.

**Figure 9 polymers-15-00437-f009:**
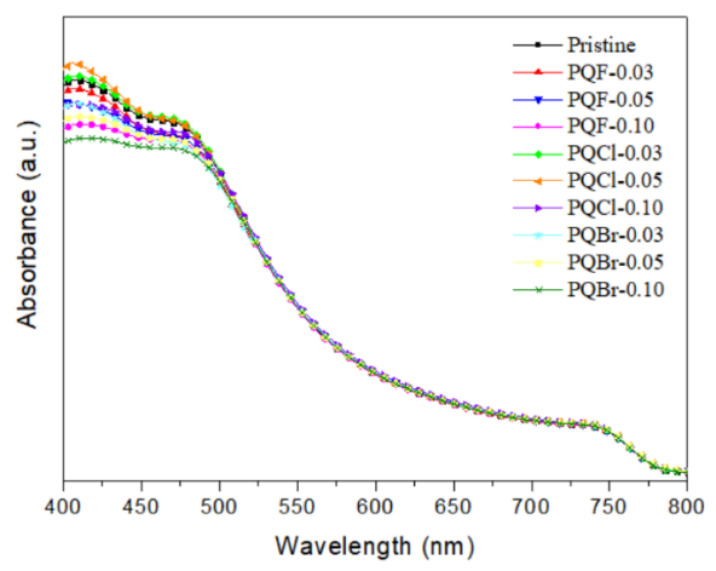
UV–Vis absorption spectra of MAPbI_3_ films deposited on cellulose derivative–modified NiO*_x_* layers.

**Figure 10 polymers-15-00437-f010:**
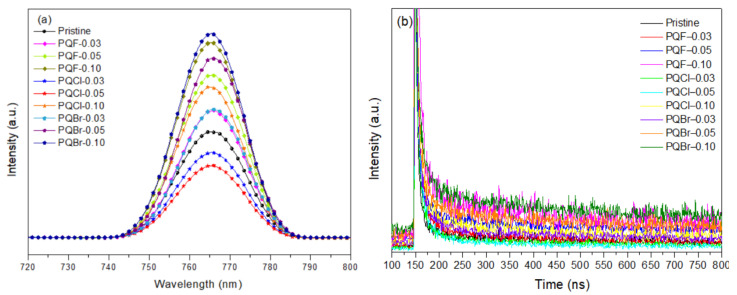
(**a**) PL and (**b**) TRPL spectra of MAPbI_3_ films deposited on cellulose derivative–modified NiO*_x_* layers (Excitation wavelength: 765 nm).

**Figure 11 polymers-15-00437-f011:**
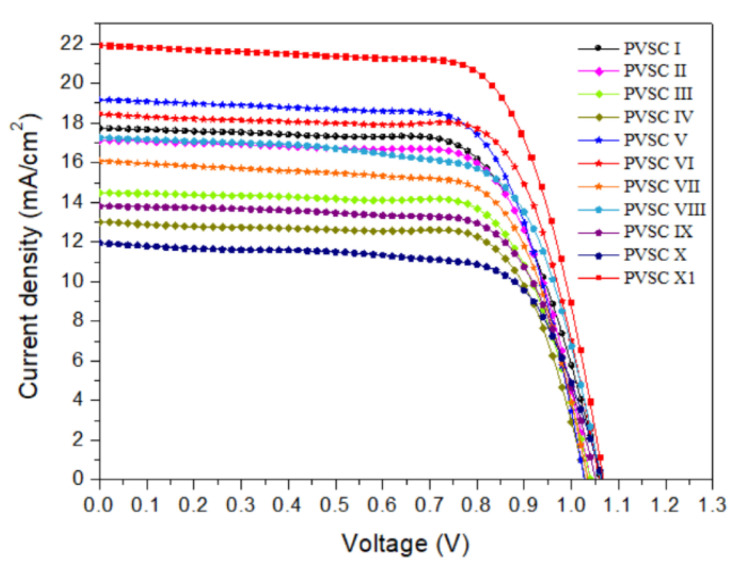
Current density–voltage characteristics of illuminated (AM 1.5G, 100 mW cm^−2^) PVSCs.

**Figure 12 polymers-15-00437-f012:**
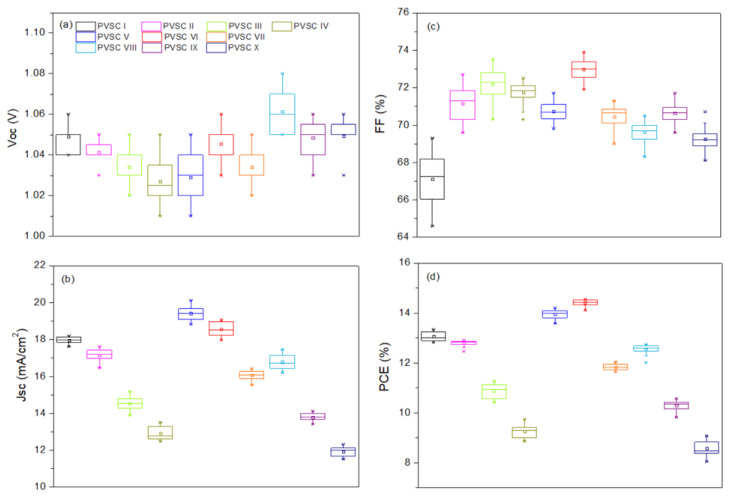
Statistical box plots for the PV parameters (**a**) *V*_OC_, (**b**) *J*_SC_, (**c**) FF, and (**d**) PCE of 20 un-encapsulated pristine and cellulose derivative (PQF, PQCl, PQBr)–modified MAPbI_3_-based PVSCs.

**Figure 13 polymers-15-00437-f013:**
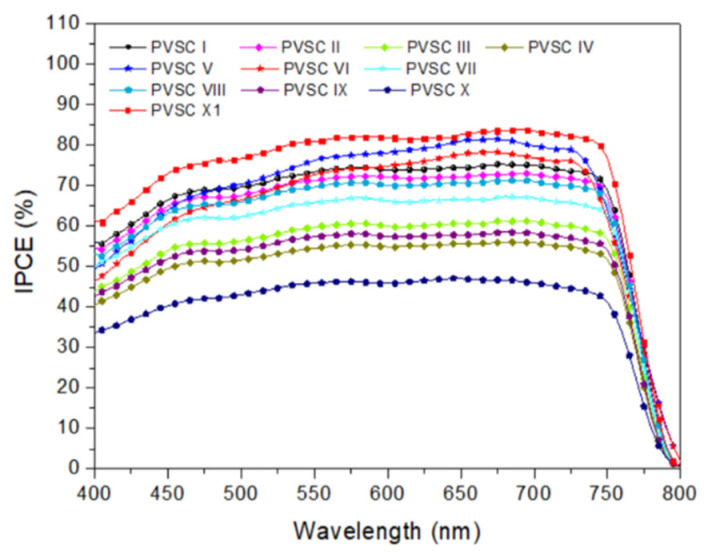
IPCE spectra of PVSCs fabricated from the cellulose derivative–modified NiO*_x_* HTLs, recorded under monochromatic irradiation.

**Figure 14 polymers-15-00437-f014:**
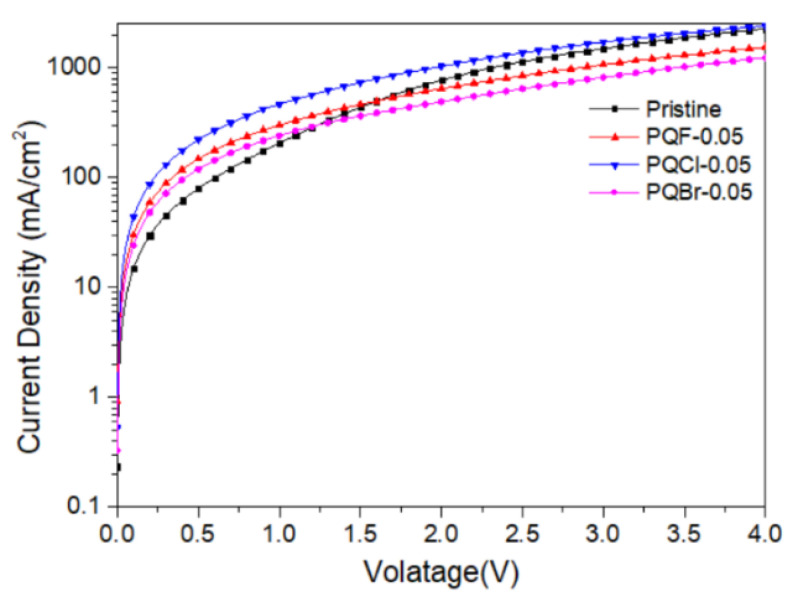
Current density–voltage plots of the hole-only devices FTO/NiO*_x_*/cellulose derivative/MAPbI_3_/Au.

**Figure 15 polymers-15-00437-f015:**
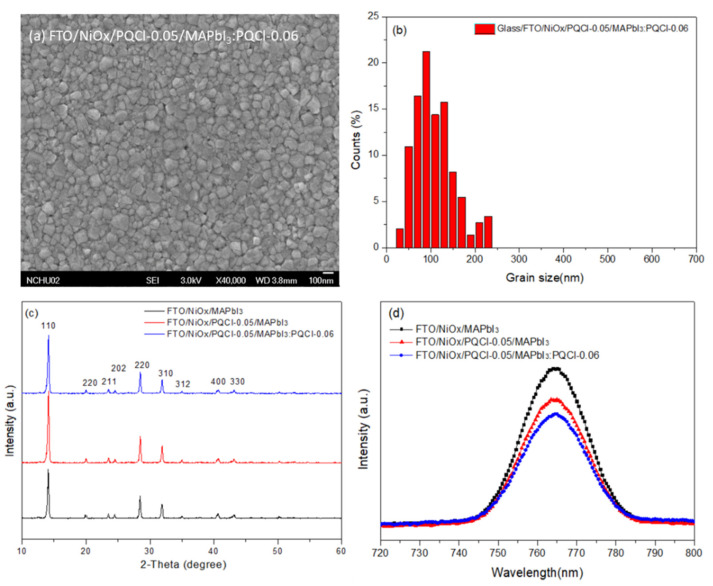
(**a**) SEM image, (**b**) MAPbI_3_ grain size distribution, (**c**) XRD pattern, and (**d**) PL spectrum of the PQCl-incorporated MAPbI_3_ film coated on the NiO*_x_* layer.

**Figure 16 polymers-15-00437-f016:**
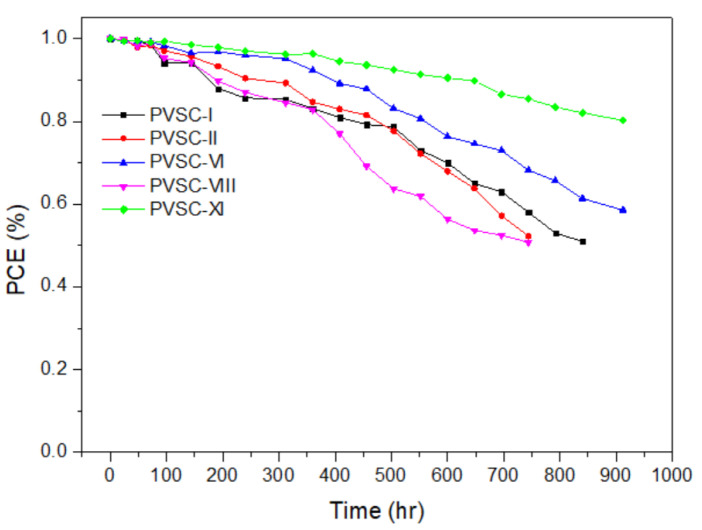
Storage-stability of PVSCs fabricated from the cellulose derivatives modified NiO_X_ HTL (measured at 30 °C and 60% relative humidity).

**Table 1 polymers-15-00437-t001:** Surface roughness, CA, and morphological characteristics of cellulose derivative–modified NiO*_x_* and MAPbI_3_ layers.

HTL	RMS * (nm)	CA ^@^ (°)	Average ^ψ^ Grain Size (nm)	RMS ^#^ (nm)	FWHM ^§^ (110)	Crystal Size ^Θ^ (nm)
NiO*_x_*	6.31 ± 0.75	38.93	244.22	9.20 ± 0.43	0.219	36.17
NiO*_x_*/PQF-0.03	10.01 ± 0.65	25.19	233.85	7.71 ± 0.25	0.220	36.01
NiO*_x_*/PQF-0.05	9.13 ± 0.78	24.51	225.26	7.39 ± 0.37	0.225	35.23
NiO*_x_*/PQF-0.10	11.26 ± 0.54	25.01	221.81	8.01 ± 0.36	0.227	34.91
NiO*_x_*/PQCl-0.03	8.84 ± 0.48	23.89	256.95	7.39 ± 0.29	0.213	37.19
NiO*_x_*/PQCl-0.05	9.41 ± 0.54	22.77	271.57	6.09 ± 0.24	0.210	37.72
NiO*_x_*/PQCl-0.10	9.46 ± 0.66	24.01	241.41	7.15 ± 0.33	0.222	35.70
NiO*_x_*/PQBr-0.03	9.99 ± 0.56	29.30	233.37	7.90 ± 0.37	0.226	35.07
NiO*_x_*/PQBr-0.05	9.31 ± 0.65	31.46	241.52	7.43 ± 0.35	0.229	34.61
NiO*_x_*/PQBr-0.10	9.96 ± 0.48	30.26	234.92	8.24 ± 0.41	0.233	34.00

* RMS: root-mean-square roughness of HTLs. ^@^ CA of cellulose derivative–modified NiO*_x_* layer. ^ψ^ Crystal grain size of MAPbI_3_, determined from SEM image. ^#^ RMS: root-mean-square roughness of MAPbI_3_ layer. ^§^ FWHM: Full width at half maximum (FWHM) of the (110) diffraction peak. ^Θ^ Crystal grain size of MAPbI_3_, determined from XRD pattern.

**Table 2 polymers-15-00437-t002:** Fitted parameters of the TRPL spectra of MAPbI_3_ films coated on cellulose derivative–modified NiO*_x_* layers coated on FTO glass.

Perovskite Layer	*A* (%)	*τ*_1_ (ns)	*B* (%)	*τ*_2_ (ns)	*τ*_avg_ (ns)
MAPbI_3_	31.73	3.92	68.27	87.45	85.74
PQF-0.03/MAPbI_3_	36.03	16.5	63.97	102.30	95.16
PQF-0.05/MAPbI_3_	32.89	6.97	67.11	160.01	156.81
PQF-0.10/MAPbI_3_	16.05	8.60	83.95	191.02	189.47
PQCl-0.03/MAPbI_3_	19.82	3.27	44.48	80.18	78.81
PQCl-0.05/MAPbI_3_	18.85	4.39	81.15	59.09	58.16
PQCl-0.10/MAPbI_3_	18.25	4.60	81.75	120.32	119.34
PQBr-0.03/MAPbI_3_	30.35	5.63	69.65	113.25	110.96
PQBr-0.05/MAPbI_3_	19.30	8.21	80.70	171.30	170.67
PQBr-0.10/MAPbI_3_	16.59	9.11	83.41	216.80	240.86

**Table 3 polymers-15-00437-t003:** PV performance data of PVSCs incorporating the cellulose derivative–modified NiO*_x_* HTLs.

PVSC	Interfacial Layer	*V*_OC_ (V)	*J*_SC_ (mA cm^−2^)	FF (%)	PCE (%)	Best PCE (%)
PVSC-Ⅰ	–	1.05 ± 0.01	17.89 ± 0.26	67.2 ± 2.1	13.08 ± 0.25	13.33
PVSC-Ⅱ	PQF-0.03	1.04 *±* 0.01	17.08 ± 0.53	71.0 ± 1.3	12.68 ± 0.21	12.89
PVSC-Ⅲ	PQF-0.05	1.04 *±* 0.01	14.49 ± 0.61	71.9 ± 1.6	10.85 ± 0.42	11.27
PVSC-Ⅳ	PQF-0.10	1.03 *±* 0.01	12.98 ± 0.51	71.4 ± 1.1	9.31 ± 0.43	9.74
PVSC-Ⅴ	PQCl-0.03	1.03 *±* 0.01	19.64 ± 0.47	70.7 ± 0.9	13.89 ± 0.30	14.19
PVSC-Ⅵ	PQCl-0.05	1.05 *±* 0.01	18.51 ± 0.53	72.9 ± 1.1	14.20 ± 0.20	14.40
PVSC-Ⅶ	PQCl-0.10	1.04 *±* 0.01	15.95 ± 0.43	70.2 ± 1.1	11.84 ± 0.20	12.04
PVSC-Ⅷ	PQBr-0.03	1.07 *±* 0.01	16.83 ± 0.62	79.7 ± 0.8	12.38 ± 0.36	12.74
PVSC-Ⅸ	PQBr-0.05	1.05 *±* 0.01	13.75 ± 0.35	70.7 ± 1.0	10.20 ± 0.38	10.58
PVSC-Ⅹ	PQBr-0.10	1.05 *±* 0.01	11.88 ± 0.37	69.4 ± 1.3	8.57 ± 0.51	9.08
PVSC-ⅩI *	PQCl-0.05	1.05 *±* 0.01	21.61 ± 0.35	70.5 ± 0.6	16.0 ± 0.53	16.53

* PVSC-XI: FTO/NiO*_x_*/PQCl-0.05/MAPbI_3_:PQCl-0.06/PC_61_BM/BCP/Ag.

## Data Availability

Not applicable.

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
