# Peer review of "Enhanced Photovoltaic Performance of Inverted Perovskite Solar Cells through Surface Modification of a NiOx-Based Hole-Transporting Layer with Quaternary Ammonium Halide–Containing Cellulose Derivatives"

_polymers, 2023, doi:10.3390/polym15020437_

Round 1

Reviewer 2 Report

The article used three quaternary ammonium halide cellulose derivatives PQF, PQCl and PQBr to modify the interfacial contact between NiOx and MAPbI3 layer and showed it reduces the interface defects and facilitates charge transport. This study also claims that the introduction of quaternary ammonium halide cellulose derivatives improves MAPbI3 grain size, crystallinity and surface roughness. Those improvements lead to an improvement in solar cell performance. The main point that PQCl improves the HTL-perovskite interface is well supported by experimental evidence. The new interface modification materials explored in this study provide valuable data points in improving the HTL-perovskite contacts, which is an important step to further enhance perovskite solar cell performance. 

However, the claim that the interface layer improves the grain size, crystallinity and surface roughness is not as well supported by the evidence. The manuscript claims the largest crystal grain size is obtained with the PQCl-0.05 sample, as shown in Table 1. In that table, the PQCl-0.05 sample has a mean grain size of 271nm compared to 244 nm for NiOx sample. However, in Figure 5, it showed a broad grain size distribution. In fact, if overlaying the distribution of the NiOx sample with PQCl-0.05, it is hard to determine if the difference in mean grain size is statistically significant. I would recommend providing statistical analysis if this study still wishes to claim that the interfacial contact increases the grain size. Similarly, for RMS, the variation in RMS between different samples is small. Statistical analysis would be recommended to claim that there is a difference due to the interfacial contact. 

This study also claims that the interface layer increases crystallinity based on pXRD results in Figure 8, where (110) diffraction peak has the highest intensity in PQCl samples. However, pXRD results are not sufficient to determine the crystallinity of thin film samples. It is more likely that the different intensity comes from the difference in the crystallographic orientation of the thin films, as the pXRD intensity is strongly affected by the preferential orientation of thin films. 

In page 7 Table 1, two crystal sizes are presented. One is from SEM results which shows crystal size around 240 nm. The other is determined from XRD patterns with grain size around 35 nm. Please clarify why there’s such a huge difference in reported grain sizes in the same sample, and which grain size should be used to draw any conclusions. 

Despite the mentioned weaknesses, the experimental data provided can support the main claim that PQCl improves the HTL-perovskite interfacial contact and boosts the performance of solar cells. I recommend Acceptance after major revisions.

Author Response

Please ee the attached file

Round 2

Reviewer 2 Report

In the first round of review it's mentioned that there is not enough statistical evidence to show a difference in the grain sizes. Because the grain size histograms in Figure 5 seem to have significant overlaps between each other, and the average grain sizes in Table 1 are relatively close in values. The authors have not provided the statistical evidence to address this concern, which could include adding an error bar to the grain size measurement, or statistical t tests. Instead, in the new version of the manuscript, a new metric FWHM of the Gaussian fit of the grain size distribution is introduced to show that the grain size FWHM is smaller in PQCl samples than the pristine samples. This metric cannot prove one sample is better than another, as comparing the standard deviation of two distributions is unrelated to comparing the averages. I'm also not able to find any literature indicating the FWHM of grain size distribution benefits the perovskite performance. Here I'd recommend either providing the statistical evidence for the grain size comparison, or dropping the grain size comparison. Since the finding that PQCl improves the HTL-perovskite interfacial contact and boosts the performance of solar cells is already worth publishing and is better supported by experimental data.

Author Response

Please see the atatched file.

Round 3

Reviewer 2 Report

The revised manuscript can be accepted.